# UMEC: Unified model and embedding compression for efficient recommendation systems

**Jiayi Shen[1], Shupeng Gui[2*], Haotao Wang[3*], Jianchao Tan[4] , Zhangyang Wang[3] , Ji Liu[4]**
[1]Texas A&M University, [2]University of Rochester, [3]University of Texas at Austin, [4]Kwai Inc.
asjyjya-617@tamu.edu,sgui2@ur.rochester.edu,{htwang,atlaswang}@utexas.edu,
{jianchaotan,jiliu}@kuaishou.com

## ABSTRACT

The recommendation system (RS) plays an important role in the content recommendation and retrieval scenarios. The core part of the system is the ranking neural network, which is usually a bottleneck of whole system performance during online inference. Hammering an efficient neural network-based recommendation system involves entangled challenges of compressing both the network parameters and the feature embedding inputs. We propose a unified model and embedding compression (**UMEC**) framework to jointly learn input feature selection and neural network compression together, which is formulated as a resource-constrained optimization problem and solved using the alternating direction method of multipliers (ADMM) algorithm. Experimental results on public benchmarks show that our **UMEC** framework notably outperforms other non-integrated baseline methods. The codes can be found at https://github.com/VITA-Group/UMEC.

## 1 INTRODUCTION

As the core component of a recommendation system (RS), recommendation models (RM) based on ranking neural networks are widely adopted in general content recommendation and retrieval applications. In general, an effective recommendation model consists of two components: a feature embedding sub-model and a prediction sub-model, as illustrated in Figure 1. Usually, an RM adopts neural networks to serve two sub-models. Formally, we denote an RM as $f(\cdot; \mathcal{W})$, where $\mathcal{W}$ is the learnable parameters of $f$. For the inference, the model $f$ takes the input feature data $\boldsymbol{x}$ to predict the confidence of the content, serving the recommendation applications. Specifically, we further define the embedding and prediction sub-models as $f_e(\cdot; \mathcal{W}_e)$ and $f_p(\cdot; \mathcal{W}_p)$ respectively, where $\mathcal{W}_e$ and $\mathcal{W}_p$ are their own learnable parameters and $\mathcal{W} = \{\mathcal{W}_e, \mathcal{W}_p\}$. The embedding feature, $\boldsymbol{v} := f_e(\boldsymbol{x}; \mathcal{W}_e)$, is the input of $f_p$ with the input data $\boldsymbol{x}$. Hence, we can express the RM as $f(\cdot; \mathcal{W}) \triangleq f_p(f_e(\cdot; \mathcal{W}_e); \mathcal{W}_p)$. Given a ranking training loss $\ell(\cdot)$ (i.e., binary cross entropy (BCE) loss), the learning goal of the ranking model can be written as

$$\min_{\mathcal{W}} \sum_{(\boldsymbol{x}, \boldsymbol{y}) \in \mathcal{D}} \ell(f(\boldsymbol{x}; \mathcal{W}), \boldsymbol{y}),$$

where $\boldsymbol{y}$ is the ground-truth label, and $\mathcal{D}$ is the training dataset.

Nowadays, extremely large-scale data have been poured into the recommendation system to predict user behavior in many applications. In the online inference procedure, the heaviest computation component is the layer-wise product between the hidden output vectors and the model parameters $\mathcal{W}$, for a neural network-based RM. A slimmed neural network structure would save a great amount of power consumption during the inference. Hence, the main idea of an RM compression is to slim the structural complexity of $f(\mathcal{W})$ and reduce the dimension of hidden output vectors.

To obtain an efficient ranking model (for example, MLP based) for an RS, one may apply existing model compression methods to MLPs directly. For example, Li et al. (2016) removes the entire filters in the network together with their connecting feature maps in terms of magnitudes, which can also

---

*Equal Contribution.

be applied to MLP structures to remove a specific neuron as well as its connections. Molchanov et al. (2019) approximates the importance of a neuron (filter) to the final loss by using the first and second-order Taylor expansions, which can also be applied to pruning the neurons without hassle. There are also some compression methods focusing on dimension reduction of embedding feature vectors. Ginart et al. (2019) proposes a mixed dimension embedding scheme by designing non-uniform embedding dimensions scaling with popularity of features. Joglekar et al. (2020) uses Neuron Input Search (NIS) to learn the embedding dimensions for the sparse categorical features.

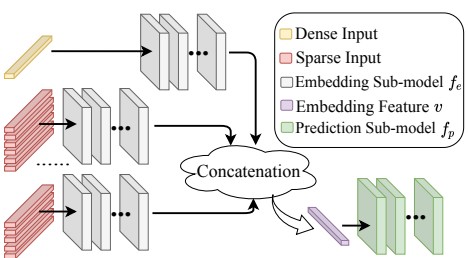

Figure 1: An example of the neural network-based recommendation system: the deep learning recommendation model (DLRM), proposed by Naumov et al. (2019).

However, the performance of these compression methods highly depends on the hyper-parameter tuning and the background knowledge of the specific recommendation model. For example, an embedding dimension compression method may require the user to search for the best dimension with multiple training and search rounds.

We would like to solve the RM compression problem with a unified resource-constrained optimization framework. It only relies on the resource consumption of the RM $f(\mathcal{W})$ to compress both the MLP and the embedding dimensions, without any multi-stage heuristics nor expensive hyper-parameters tuning. Our novel unified model and embedding compression (**UMEC**) framework directly satisfies both the requirement of resource consumption of the ranking neural network model and the prediction accuracy goal of the ranking model, with end-to-end gradient-based training. We reformulate the optimization of training loss associated with hard constraints into a minimax optimization problem solved by the alternating direction method of multipliers (ADMM) method (Boyd et al., 2011).

To summarize our contributions, we present the merits of **UMEC** as the following:

- To the best of our knowledge, **UMEC** is the first unified optimization framework for the recommendation system scenario. Unlike those existing works that treat the selection of input feature and compression of the model as two individual problems, **UMEC** jointly learns both together via unifying both original prediction learning goal and the model compression related hard constraints.

- We reformulate the joint input feature selection and model compression task as a constrained optimization problem. We convert resource constraints and $L_0$ sparsity constraints into soft optimization energy terms and solve the whole optimization using ADMM methods.

- Extensive experiments performed over large-scale public benchmarks show that our method largely outperforms previous state-of-the-art input feature selection methods and model compression methods, endorsing the benefits of the proposed end-to-end optimization.

## 2 RELATED WORK

**Recommendation Models** With the recent development of deep learning techniques, plenty of works have proposed learning-based recommendation systems to grapple with personalized recommendation tasks. Hidasi et al. (2015) applies recurrent neural networks (RNN) on long session data to model the whole session and achieve better accuracy than the traditional matrix factorization approaches. Covington et al. (2016) proposes a high-level recommendation algorithm for YouTube videos by specifying a candidate generation model followed by a separate ranking model, which demonstrates a significant performance boost thanks to neural networks. Wu et al. (2019) proposes a recommendation algorithm based on the graph neural network to better capture the complex transitions among items, achieving a more accurate item embedding. Their method outperforms traditional sequential approaches. Naumov et al. (2019) exploits categorical features and designs a parallelism scheme for the embedding tables to alleviate the limited memory problem. Huang et al. (2020) proposes a multi-attention based model to mitigate the low efficiency problem in the group recommendation scenarios.

**Input Feature Compression for Recommendation Models**   The numerous input features for practical recommendation scenarios necessitate the selection of useful ones to save memory resources and facilitate computational efficiency during inference. In factorization machines(FM)-based and context-aware recommendation systems, for example, Mao et al. (2017) conducts input feature selection in an automatic manner by feature ranking and feature sub-sampling, while managing to improve the prediction quality. Ginart et al. (2019) designs a mixed dimension embedding layers scheme and adjusts the dimension of a particular embedding feature according to the frequency of that item. Joglekar et al. (2020) performs Neural Input Search (NIS) method to search for the embedding sizes for the sparse categorical features, and designs the multi-size embedding framework to leverage the model capacity more efficiently. Song et al. (2020) proposes a discovering framework to automatically find the interaction architecture of the prediction model.

**Model Compression**   Model compression aims to reduce the complexity of Deep Neural Network (DNN) models to achieve inference efficiency and resource saving. Generally, model compression techniques can be categorized into pruning (Han et al., 2015; Wen et al., 2016; Molchanov et al., 2019), quantization (Hubara et al., 2017; Wu et al., 2018; Ajanthan et al., 2019) and distillation (Hinton et al., 2015; Polino et al., 2018; Tung & Mori, 2019). Han et al. (2015) proposes a magnitude-based and element-wise pruning technique, which does not guarantee the reduction of computation efficiency, energy and memory costs. Structured pruning can solve this issue by removing whole certain types of structures from the network, such as filter pruning and channel pruning methods (Li et al., 2016; He et al., 2017; Molchanov et al., 2019). Recent works (Frankle & Carbin, 2019; Chen et al., 2020b;a; 2021c;b;a; Ma et al., 2021) reveal that with appropriate compression techniques, the identified sparse subnetworks are capable of training in isolation to match the dense model performance. In the meanwhile, researchers also develop great interest in investigating hardware-aware compression, which utilizes practical resource requirements(e.g., energy, latency, Flops) to guide the compression (He et al., 2018; Yang et al., 2019; Liu et al., 2019).

## 3   Unified Model and Embedding Compression on RS

For an arbitrary NN-based recommendation model $f(\cdot)$ in an RS, the terminology of "compression" refers to reduce the power consumption with respect to all the computation operations for the inference. The learning parameters of $f(\mathcal{W})$ can be expressed as $\mathcal{W} := \left\{ \boldsymbol{W}^{(l)} | l \in [L] \right\}$, where $L$ is the number of layers in the $f(\mathcal{W})$. For an arbitrary layer $l$, the layer's weight is denoted as $\boldsymbol{W}^{(l)} \in \mathbb{R}^{d_{\text{out}} \times d_{\text{in}}}$. The recommendation inference on the layer $l$ is executed as

$$\boldsymbol{x}^{(l+1)} = \sigma \left( \boldsymbol{W}^{(l)} \boldsymbol{x}^{(l)} + \boldsymbol{b}^{(l)} \right),$$

where $\boldsymbol{x}^{(l)} \in \mathbb{R}^{d_{\text{in}}}$ is the input of $l$-th layer, and $\boldsymbol{b}^{(l)}$ is the corresponding learning bias term. Since the RM consists of two components: feature embedding sub-model and neural network prediction sub-model, the "compression" can be interpreted as shrinking the dimension of embedding feature $\boldsymbol{v}$ and reducing the number of parameters in $\mathcal{W}$ and $\boldsymbol{b}$.

**Perspective of model parameters $\mathcal{W}$ and $\boldsymbol{b}$**   It is to reduce the input or output neurons for the specific layer $l$. For example, the $i$-th neuron of $\boldsymbol{x}^{(l+1)}$ and $j$-th neuron of $\boldsymbol{x}^{(l)}$ are chosen to remove, then the weight matrix $\boldsymbol{W}^{(l)}$ will be reshaped to $(d_{\text{out}} - 1) \times (d_{\text{in}} - 1)$.

**Perspective of feature embedding vectors $\boldsymbol{v}$**   The reduction of $d_{\text{in}}$ would result in the elimination of the input features. For example, $\boldsymbol{v}^{(1)}$ is defined as the concatenation of a series of embedding vectors, $\boldsymbol{v}^{(1)} = [\boldsymbol{e}_1; \boldsymbol{e}_2; \cdots ; \boldsymbol{e}_n]$, where $[\cdot; \cdot; \cdot]$ represents the concatenation operator, and $n$ denotes the number of embedding feature vectors in the RS. Without loss of generality, for the first embedding vector $\boldsymbol{e}_1 \in \mathbb{R}^{d_1}$, the zero-out of first $d_1$ columns in $\boldsymbol{W}^{(1)}$ will result in the elimination of the whole embedding vectors for $\boldsymbol{e}_1$. On the other hand, the compression on $f_e(\mathcal{W}_e)$ would reduce the feature embedding vectors' dimensions.

### 3.1   A compression example: Group Lasso for Weight Pruning

We first consider a simple example with an RS model consisting of $f_e(\mathcal{W}_e)$ and $f_p(\mathcal{W}_p)$. For this RS, to reduce the computation consumption, we present a straightforward model compression method

with Group Lasso. The method considers both the feature embedding selection and network slimming. We only need to apply the sparse regularization terms to the weight matrices. The optimization problem can be expressed as

$$\min_{\mathcal{W}} \ell(\mathcal{W}) + \lambda_1 \sum_i \left\| \boldsymbol{W}^{(1)}_{:,g_i} \right\|_1 + \sum_{l=2}^{L} \lambda_l \sum_i \left\| \boldsymbol{W}^{(l)}_{:,i} \right\|_1 , \tag{1}$$

where $\ell(\mathcal{W})$ is the original objective function . The $\lambda$s are hyper-parameters to control the importance of these regularization terms. $g_\cdot$ denotes the feature groups for the input layer. $\| \cdot \|_1$ indicates the $\mathcal{L}_1$ norm. The optimization problem (1) restricts the group-wise sparsity for the input layer and the neuron-wise sparsity (or group size = 1) for the rest model layers while training with the original objective function $\ell(\mathcal{W})$. The optimization problem can be solved with the projected gradient descent method. However, the optimization problem (1) has its own limitation. The hyper-parameters $\lambda$s only have the latent impact on the model size and we could not directly affect the model size. It is not friendly when the users would like to restrict the model size to a specific number. On the other hand, the hyper-parameters also make the optimization problem hard to tune, requiring multiple arguments searching rounds. To avoid these shortages, we are supposed to have new approaches to handling the compression of RS.

## 3.2 RESOURCE-CONSTRAINED MODEL COMPRESSION

In this section, we propose a novel unified model and embedding compression method on RS and treat it as a constrained optimization problem. We discuss how to solve the problem using a gradient-based optimization algorithm.

**Computation Resource Function for RS Model**  To avoid the hyper-parameter tuning in the optimization of problem (1), we define the resource function to measure the resource consumption for a specific RS model structure, which can be used to guide the compression of structure. For example, if considering the computation Flops, the resource of an MLP can be defined as

$$R_{\text{Flops}} = \sum_{l=1}^{L} \left( 2 \times d_{\text{in}}^{(l)} \times d_{\text{out}}^{(l)} + d_{\text{out}}^{(l)} \right),$$

where $d_{\text{in}}^{(l)}$ and $d_{\text{out}}^{(l)}$ indicate the number of input and output neurons. For a pruned layer $l$, the input and output dimension are restricted by the number of pruned neurons, annotated as $s^{(l)}$ and $s^{(l+1)}$. Therefore, we can define a resource consumption function in terms of $s$ from each layer as

$$R_{\text{Flops}}(\boldsymbol{s}) \triangleq \sum_{l=1}^{L} \left( 2 \times \left( d_{\text{in}}^{(l)} - s^{(l)} \right) \times \left( d_{\text{out}}^{(l)} - s^{(l+1)} \right) + \left( d_{\text{out}}^{(l)} - s^{(l+1)} \right) \right).$$

For simplicity, the resource function $R_{\text{Flops}}$ only takes the $s$ as the number of neurons. In a more general case, $s$ can be substituted with the product between the number of pruned groups and the size of each pruned group.

**Resource-Constrained RS Model Compression**  With the definition of the resource function, we can formulate the following optimization problem,

$$\min_{\mathcal{W}} \ell(\mathcal{W}), \quad \text{s.t. } R_{\text{Flops}}(\mathcal{W}) \leq R_{\text{budget}}, \tag{2}$$

where $R_{\text{budget}}$ denotes the upper bound of the overall model computation Flops set by users. Since we consider the group-wise sparsity for each layer, we utilize the set of variables $\boldsymbol{s} := \{s^{(1)}, s^{(2)}, \ldots, s^{(L)}\}$ to control the model's Flops. The $s^{(l)}$ defines the lower bound of the $l$-th layer's

zero group number for the pruned model. Therefore, we reformulate the optimization problem (2) as

$$\min_{\mathcal{W}, \boldsymbol{s}} \ell(\mathcal{W}) \tag{3a}$$

$$\text{s.t. } R_{\text{Flops}}(\boldsymbol{s}) \leq R_{\text{budget}} \tag{3b}$$

$$\sum_i \mathbb{I} \left( \left\| \boldsymbol{W}_{:,i}^{(l)} \right\|_2^2 = 0 \right) \geq s^{(l)}, \, \forall \, l = 2, \dots, L, \tag{3c}$$

$$\sum_i \mathbb{I} \left( \left\| \boldsymbol{W}_{:,g_i}^{(1)} \right\|_2^2 = 0 \right) \geq s^{(1)}, \tag{3d}$$

where $\mathbb{I}(\cdot)$ is an indicator function with the output $\{0, 1\}$. When the condition is satisfied, the indicator function results in a value of 1, otherwise, it would be 0. $\sum_i \mathbb{I} \left( \left\| W_{:,i}^{(l)} \right\|_2^2 = 0 \right)$ denotes the zero group number for the $l$-th layer. The resource constraint (3b) bounds the resource consumption of the recommendation model. The inequalities in both (3c) and (3d) regularize the Flops consumption for each layer.

## 3.3 Optimization

To address the optimization problem (3), we adopt the alternating direction method of multipliers (ADMM) for the reformulation. In details, the optimization problem contains non-continuous indicator function in constraint (3c, 3d), and non-convex constraint (3b), which makes the problem difficult to solve. Therefore, we first reformulate the inequality constraints as soft regularizations and introduce minimax optimization with dual variables. Then we tackle the non-differentiable objective function using self-defined numerical differentiation. At last, we summarize all the optimization details into a gradient-based optimization formulation.

**Minimax Optimization Reformulation**   Let $\|\boldsymbol{W}_{:,g}\|_{s,2}$ be the bottom-$(s, 2)$ "norm", which denotes the Frobenius-norm of the sub-matrix (group) composed of bottom-$s$ groups, by sorting from top to bottom in terms of norm of each group. The equivalent formulation of the sparsity constraint (3c, 3d) can be

$$\|\boldsymbol{W}_{:,g}\|_{s,2}^2 = 0 \Leftrightarrow \sum_i \mathbb{I} \left( \|\boldsymbol{W}_{:,g_i}\|_2^2 = 0 \right) \geq s. \tag{4}$$

The above equivalence (4) is proved by Tono et al. (2017), where the LHS is called DC (difference of convex functions) representation of the $\mathcal{L}_0$-constraint. With the transformation, we can reformulate the problem in Eq. (3), into this minimax optimization problem:

$$\min_{\mathcal{W}, \boldsymbol{s}} \max_{\boldsymbol{y}, z \geq 0} \ell(\mathcal{W}) + \underbrace{y^{(1)} \left\| \boldsymbol{W}_{:,g}^{(1)} \right\|_{\lceil s^{(1)} \rceil, 2}^2 + \sum_{l=2}^{L} y^{(l)} \left\| \boldsymbol{W}_{:,:}^{(l)} \right\|_{\lceil s^{(l)} \rceil, 2}^2}_{\text{sparsity loss: } \mathcal{S}(\boldsymbol{y}, \boldsymbol{s}, \mathcal{W})} + \underbrace{z \left( R_{\text{Flops}}(\boldsymbol{s}) - R_{\text{budget}} \right)}_{\text{resource loss}}, \tag{5}$$

where $y$ and $z$ are dual variables. The operator $\lceil \cdot \rceil$ indicates the ceiling function. In the problem (5), the importance of regularization terms are tuned automatically based on the minimax optimization. The sparsity loss $\mathcal{S}(\boldsymbol{y}, \boldsymbol{s}, \mathcal{W}) := y^{(1)} \left\| \boldsymbol{W}_{:,g}^{(1)} \right\|_{\lceil s^{(1)} \rceil, 2}^2 + \sum_{l=2}^{L} y^{(l)} \left\| \boldsymbol{W}_{:,:}^{(l)} \right\|_{\lceil s^{(l)} \rceil, 2}^2$ and the resource loss $z \left( R_{\text{Flops}}(\boldsymbol{s}) - R_{\text{budget}} \right)$ are introduced to substitute the original constraints (3b, 3c, 3d). Therefore, the optimal solution to the new problem (5) can serve as the optimal for the original problem (3). To solve it, we present the details of update rules for all learnable variables in appendix A.1.

We wrap up all the update rules together to be a unified optimization algorithm, as described in Algorithm 1. In this algorithm, we only show an example that the input layer is column-wise grouped. For a more general case, the column-wise group sparsity can be applied to an arbitrary layer in the RS model.

**Prune and Finetune**   Solving the minimax optimization problem (5) will make some channel weights extremely close but not exactly equal to zeros. In order to extract the slimmed network, for

each $\boldsymbol{W}^{(l)}$, we discard the $s^{(l)}$ groups with smallest $\mathcal{L}_2$ norms $\|\boldsymbol{W}^{(l)}_{.,g_i}\|_2$ (extremely close to 0) after we have solved problem (5). Then we finetune the explicitly pruned network to further improve the performance.

---

**Algorithm 1:** Gradient-based algorithm to solve problem (5) for **UMEC**.

---

**Input:** Resource budget $R_{\text{budget}}$, learning rates $\eta_1, \eta_2, \eta_3, \eta_4$, number of total iterations $\tau$.
**Result:** DNN pruned weights $\mathcal{W}^*$.

1   Initialize $t = 1, \mathcal{W}^1$ ;                 `// random or a pre-trained dense model`
2   **for** $t \leftarrow 1$ **to** $\tau$ **do**
3     $\mathcal{W}^{t+1} = \text{Prox}_{\eta_1 \mathcal{S}(\boldsymbol{y}^t, \boldsymbol{s}^t, \mathcal{W}^t)} \left( \mathcal{W}^t - \eta_1 \hat{\nabla}_{\mathcal{W}} \ell \left( \mathcal{W}^t \right) \right)$ ;         `// Proximal-SGD`
4     $\boldsymbol{s}^{t+1} = \boldsymbol{s}^t - \eta_2 \left( \tilde{\nabla}_{\boldsymbol{s}} \mathcal{S} \left( \boldsymbol{y}^t, \boldsymbol{s}^t, \mathcal{W}^{t+1} \right) + \tilde{\nabla}_{\boldsymbol{s}} z^t \left( R_{\text{Flops}} \left( \boldsymbol{s}^t \right) - R_{\text{budget}} \right) \right)$ ; `// Gradient (STE)`
     Descent
5     $y^{(1)^{t+1}} = y^{(1)^t} + \eta_3 \left( \left\| \boldsymbol{W}^{(1)^{t+1}}_{.,g} \right\|^2_{\lceil \boldsymbol{s}^{(1)^{t+1}} \rceil, 2} \right)$ ;            `// Gradient Ascent`
6     $y^{(l)^{t+1}} = y^{(l)^t} + \eta_3 \left( \left\| \boldsymbol{W}^{(l)^{t+1}} \right\|^2_{\lceil \boldsymbol{s}^{(l)^{t+1}} \rceil, 2} \right), \forall \, l = 2, \cdots, L$
     $z^{t+1} = z^t + \eta_4 \left( R_{\text{Flops}} \left( \boldsymbol{s}^t \right) - R_{\text{budget}} \right)$ ;           `// Gradient Ascent`
7   $\mathcal{W}^* = \mathcal{W}$

---

## 4   EXPERIMENTS

To demonstrate the effectiveness of **UMEC**, we run experiments on a standard public RM over the real-world recommendation dataset. And we would like to answer the following questions: *Whether our unified optimization framework that jointly considers input feature selection and prediction model compression has superiority over previous state-of-the-art RS compression methods which solve the problem from only one aspect? If yes, why?*

### 4.1   EXPERIMENTAL SETUP

**Dataset and Model**   We conduct all our experiments on the state-of-the-art ads CTR prediction model named DLRM (Naumov et al., 2019), as illustrated in Figure 1. Following Naumov et al. (2019), we use the Criteo AI Labs Ad Kaggle[1] and Terabyte[2] datasets for our experiments. The Criteo AI Labs Ad Kaggle dataset contains approximately 45 million click log samples collected over seven days. The Terabyte dataset consists of approximately 4.4 billion click log samples collected over 24 days and we perform uniform sampling with 12.5% sampling rate from the raw data following the sampling scheme in Naumov et al. (2019). Both datasets contain samples that have 13 continuous and 26 categorical input features. Following the official setting, for both datasets we split the data of the last day into validation and testing sets, and use data from the rest days as the training set.

**Baseline methods**   The latency of an ads CTR prediction model mainly comes from two sources: feature embedding and prediction model inference. Feature embedding layers map categorical input values into a vector space. Both the number of input categorical values and the output feature space dimension can affect the latency of feature embedding stage. The inference latency of the predication model mainly depends on the prediction model size. Following the above analyses, methods to accelerate ads CTR prediction models mainly fall into three categories:

- Prediction model compression: compressing the prediction model with traditional model compression methods, such as Li et al. (2016); Molchanov et al. (2019); Yang et al. (2019).

- Input feature selection: reducing the number of categorical input features by filtering out non-essential ones.

---

[1]https://www.kaggle.com/c/criteo-display-ad-challenge
[2]https://labs.criteo.com/2013/12/download-terabyte-click-logs/

- Embedding dimension reduction: shrinking the dimension of embedding layers' output space. Two recent works (Ginart et al., 2019; Joglekar et al., 2020) fall into this category.

Our unified optimization framework jointly considers input feature selection and model compression. We will further show that our method can be generalized to considering input feature dimension-reduction as well. So we compare our method with baselines from all three categories to show the superiority of our method over those baselines tackling the RS compression problem from only one aspect.

**General Training and Evaluation Details**  We use the following settings for our experiments on the Criteo Ad Kaggle dataset: we use SGD optimizer with learning rate 0.1 to optimize the BCE loss ($\ell(\mathcal{W})$ in our method); we set training batch size to 128, initial feature embedding dimension to 16, and use the three-layer MLP prediction model with hidden dimensions 512 and 256.

For our method, we choose Adam optimizer (Kingma & Ba, 2014) for $s$ and SGD optimizer for $\mathcal{W}$, $y$, and $z$. We set learning rates $\eta_1, \eta_2, \eta_3, \eta_4$ in Algorithm 1 to be $[0.1, 0.05, 0.1, 2.0]$ respectively, and set $\tau$ to be 306,969 which equals to 1 epoch. In our experiments, we initialize the model weights from a dense model that has been pre-trained for 1 epoch using $\ell(\mathcal{W})$ to facilitate training. After the training process in Algorithm 1, we zero out the redundant neurons (as described in the last paragraph in Section 3) and extract the sub-network as the pruned model. We then perform fine-tuning for 2 epochs. For a fair comparison, we also fine-tune 2 epochs for all baseline methods. Following Naumov et al. (2019), the best accuracy on the validation set is reported.

For experiments on the Criteo Ad Terabyte dataset, the training settings generally follow those mentioned above with several exceptions: batch size equal to 4096 , $\tau$ equal to 157,655, the hidden dimensions of the three-layer MLP prediction model equal to 256 and 128.

**Evaluation Metrics**  We evaluate the performance with the following two metrics: *Accuracy* and *Compression Ratio (CR)*. **Accuracy:** CTR prediction accuracy on the validation set. **Compression ratio (CR):** the ratio between the Flops that has been deducted and original model Flops. More formally, $\text{CR} = 1 - p_f(\mathcal{W})/p_f(\mathcal{W}_0)$, where $p_f(\cdot)$ calculates the Flops of a (sub-)model. $\mathcal{W}$ and $\mathcal{W}_0$ are the pruned and original dense prediction models, respectively. The larger CR, the more compact model we will obtain.

## 4.2 PREDICTION MODEL COMPRESSION FOR RS MODEL

We design and conduct extensive experiments to examine the compression performance among our method and several state-of-the-arts model compression methods. The baseline methods include handcrafted structures, one-shot magnitude pruning (MP) (Li et al., 2016), and two recent state-of-the-art methods, namely Taylor pruning (TP) (Molchanov et al., 2019) and energy-constrained model compression (ECC) (Yang et al., 2019).

In Figure 2 and 3, we demonstrate the best accuracy on validation set under different compression ratios (CRs) on the Criteo AI Labs Ad Kaggle and Terabyte datasets. Each curve shows the CR-Accuracy trade-off of one compression method. The doted magenta line represents the accuracy achieved by the original dense DLRM model.

Several observations can be drawn from this comparison figure. First, our method continuously outperforms all other methods with a margin of at least 0.01%. Second, only our method can successfully compress the original dense model to the ideal budget while still manages to maintain the performance without any degradation within a certain CR scope.

These results show that our joint model compression and input selection method can achieve much better balance of efficiency vs accuracy than traditional model compression methods.

## 4.3 INPUT FEATURE SELECTION FOR RS MODEL

We evaluate the effectiveness of our proposed method on input feature selection task. The input feature selection task can be easily realized using our proposed framework, by keeping the resource constraint for the features groups in input layer (*i.e.*, Eq. (3c)) and ignoring constraint for weights of hidden layers(*i.e.*, Eq. (3d)). We use Group Lasso as a self-designed baseline. The loss function

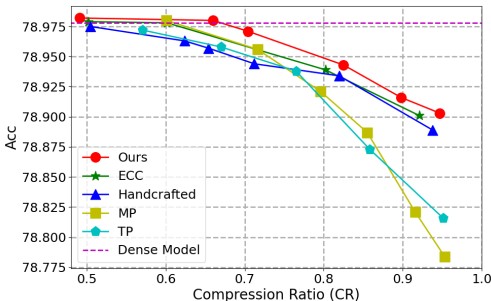 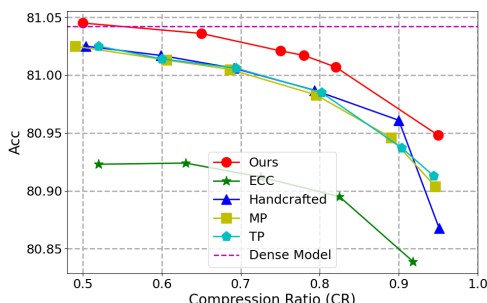

Figure 2: Results on RS prediction model compression by different methods on the Criteo AI Labs Ad Kaggle dataset.

Figure 3: Results on RS prediction model compression by different methods on the Criteo AI Labs Ad Terabyte dataset.

of Group Lasso is shown in Eq. (1). Since we are only considering input feature selection in this sub-section, the sparsity regularization on hidden layers (*e.g.*, the second term in $\mathcal{S}(\boldsymbol{y}, \boldsymbol{s}, \mathcal{W})$) is ignored. We tune hyper-parameter $\lambda_1$ among 0.0005, 0.001, 0.005, 0.01 for the Criteo AI Labs Ad Kaggle dataset, and 0.001, 0.003, 0.005 for the Terabyte dataset, in order to get different compression ratios. After Group Lasso training, all input features whose $\mathcal{L}_2$ norms are below a threshold $th$ will be removed, and the Flops for the new model are calculated accordingly. We empirically find that setting $th$ to 0.01 on the Kaggle dataset and 0.001 on the Terabyte dataset can achieve good performance. For both our method and Group Lasso, we train and fine-tune for 1 epoch and 2 epochs, respectively. In Figure 4 and 5, the performance on the holdout validation set at the end of the compressing

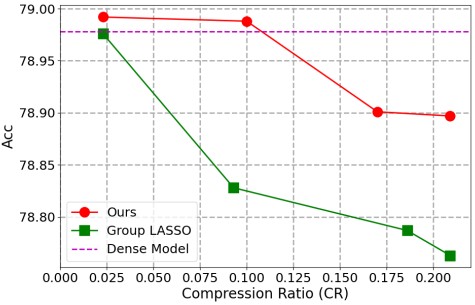 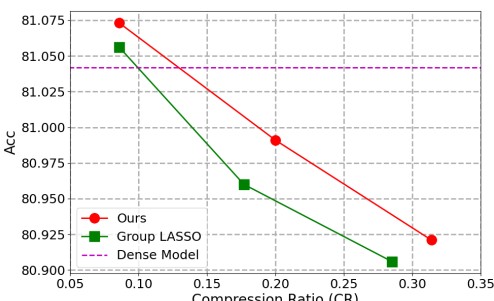

Figure 4: Results on RS input feature selection by different methods on the Criteo AI Labs Ad Kaggle dataset.

Figure 5: Results on RS input feature selection by different methods on the Criteo AI Labs Ad Terabyte dataset.

process is reported. We demonstrate the effectiveness of our method in input feature selection task, by showing its superior performance under all the compress ratios. Another observation is that the pruned model of both methods will fail to recover the performance of the original dense model if too many input features are removed. For example, at a 0.21 compression ratio on the Kaggle dataset, which corresponds to removing 9 out of the total 27 input features, both methods have a considerable drop of accuracy. This indicates that reducing the number of input features will cause considerable harm to the model's performance.

## 4.4 EMBEDDING DIMENSION REDUCTION FOR RS MODEL

We show that our method can also be applied to embedding dimension reduction. In this new scenario, $\mathcal{W}$ in Eq. (3) is embedding layer weights, instead of previous prediction model weights. Eq. (3d) is removed (*i.e.*, no longer to prune the input dimension). The rest parts of the objective function remain the same.

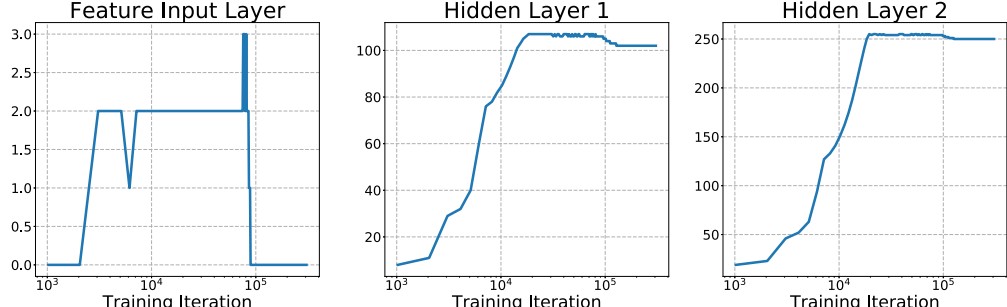

Figure 6: The number of pruned features/neurons for each layer during the training process using **UMEC** with $R_{\text{budget}} = 0.5 \times p_f(\mathcal{W}_0)$ on the Criteo AI Labs Ad Kaggle dataset, where $p_f(\mathcal{W}_0)$ denotes the Flops of the original dense model. For the input layer, the mentioned dense model has 27 features in total. For the two hidden layers, it has 512 and 256 neurons respectively. The convergency of the binary cross entropy (BCE) loss during training is shown in Figure 7 in the appendix.

We compare our method with the state-of-the-art embedding dimension reduction method named mixed dimension embedding layers (MDEL) (Ginart et al., 2019). Following Ginart et al. (2019), we use size of the embedding layers to help define compression ratio. More specifically, in this setting, $\text{CR} = 1 - p_s(\mathcal{W})/p_s(\mathcal{W}_0)$, where $p_s(\cdot)$ calculates the number of parameters in a (sub-)model. $\mathcal{W}$ and $\mathcal{W}_0$ are the pruned and original dense embedding sub-models, respectively. To evaluate performance change after pruning, we define $\Delta_{acc} = \text{Acc}_{\mathcal{W}} - \text{Acc}_{\mathcal{W}_0}$, where $\text{Acc}_{\mathcal{W}}$ and $\text{Acc}_{\mathcal{W}_0}$ are CTR prediction accuracy of a pruned and original dense model, respectively.

MDEL only provides results on the Criteo AI Labs Ad Kaggle dataset, so we conduct the comparison on this single dataset. Experiment results are shown in Table 1. As we can see, both methods have slight improvement in accuracy at around 50% CR, while the accuracy improvement of our method is larger than that of MDEL. Also, our method can still achieve better accuracy improvements than MDEL with even 10% larger CR. These results show that our ADMM based compression algorithm is better than MDEL.

Table 1: Results on embedding dimension reduction.

| Method | CR | $\Delta_{acc}$ |
|--------|-----|------|
| MDEL | 0.5 | 0.10% |
| Ours | 0.5 | 0.24% |
| | 0.6 | 0.16% |

### 4.5 SPARSITY ANALYSES OF **UMEC**

Below we present how the sparsity of the prediction model evolves during the pruning phase in Section 4.2. As a recall, the aforementioned prediction model consists of one input layer and two hidden layers, with 27 input features, 512 and 256 neurons respectively. We take the Criteo AI Labs Ad Kaggle dataset as an example. As shown in Figure 6, when $R_{\text{budget}} = 0.5 \times p_f(\mathcal{W}_0)$, the pruned input features/neurons converge to 0, 102 and 250 respectively. During training, the last hidden layer gets the most percentage of neurons removed and the input layer gets the least percentage. We can come to several conclusions from this result. First, such phenomenon aligns with the observation in Section 4.3 and serves as a piece of evidence that we need enough input features to adequately represent the raw input information and contribute to the downstream classification task. Second, to perform the relatively easier task, i.e., binary classification, the requirement of neurons from the last hidden layer falls far below the original design. Thus considerable resource consumption can be saved here. Third, it demonstrates that our resource-constrained unified optimization plays a smart role in globally finding the optimal resource allocations across all layers.

## 5 CONCLUSION

The goal of this paper is to tackle with the new problem that jointly compressing input features and neural network structures in the recommendation models. We propose **UMEC** framework by integrating these two objectives into one unified constrained optimization problem, solved by the ADMM method. We conduct extensive experiments and demonstrate the effectiveness of our proposed **UMEC** method by observing its superior performance than other state-of-the-art baseline methods.

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

## A APPENDIX

### A.1 UPDATE RULE FOR ALL LEARNABLE VARIABLES

To solve the optimization problem (5), we utilize the ADMM strategy to solve a series of sub-problems as follows.

**Update** $\mathcal{W}$ The optimization problem on $\mathcal{W}$ can be solved with proximal-SGD (Nitanda, 2014). The definition of the proximal operator is as follows:

$$\text{Prox}_{\eta \mathcal{S}(\boldsymbol{y}, \boldsymbol{s}, \mathcal{W})}(\bar{\mathcal{W}}) = \arg\min_{\mathcal{W}} \frac{1}{2} \left\| \mathcal{W} - \bar{\mathcal{W}} \right\|^2 + \eta \mathcal{S}(\boldsymbol{y}, \boldsymbol{s}, \mathcal{W})$$

where $\bar{\mathcal{W}}$ is the direct stochastic gradient descent update of $\mathcal{W}$. The sub-problem above has the closed-form solution for $\mathcal{W}^* := \left\{ \boldsymbol{W}^{(1)*}, \boldsymbol{W}^{(2)*}, \ldots, \boldsymbol{W}^{(L)*} \right\}$:

$$\boldsymbol{W}_i^* = \begin{cases} \bar{\boldsymbol{W}}_i, & \text{if } \left\| \bar{\boldsymbol{W}}_i \right\| >= \left\| \bar{\boldsymbol{W}}_{\text{least-}\lceil s \rceil} \right\|, \\ \frac{1}{1+2\eta y} \bar{\boldsymbol{W}}_i, & \text{otherwise,} \end{cases}$$

where $\boldsymbol{W}_i$ denotes the $i$-th sub-matrix, and $\boldsymbol{W}$ is the weight for an arbitrary layer.

**Update dual variables** $z$ **and** $y$ For the sub-optimization problem, we have

$$\max_{\boldsymbol{y}, z \geq 0} \; y^{(1)} \left\| \boldsymbol{W}_{\cdot, g}^{(1)} \right\|_{\lceil s^{(1)} \rceil, 2}^2 + \sum_{l=2}^{L} y^{(l)} \left\| \boldsymbol{W}_{\cdot\cdot}^{(l)} \right\|_{\lceil s^{(l)} \rceil, 2}^2 + z \left( R_{\text{Flops}}(\boldsymbol{s}) - R_{\text{budget}} \right),$$

and we can adopt the gradient ascent method as the update rule.

**Update** $\boldsymbol{s}$ The optimization on $\boldsymbol{s}$ relies on both sparsity and resource loss. However, both of them are non-differentiable due to the ceiling function $\lceil s \rceil$. Straight-through estimator (STE) (Bengio et al., 2013) is an effective approach for optimization through the non-differentiable functions. The main idea is to adopt some simple proxy to be the derivative of the non-differentiable formulation. Thus, the back-propagation can be applied as what happened in the differentiable case. For $\|\boldsymbol{W}\|_{s,2}^2$, we can apply the numerical differentiation $\|\boldsymbol{W}\|_{s+1,2}^2 - \|\boldsymbol{W}\|_{s,2}^2$ as the proxy derivative of $\|\boldsymbol{W}\|_{s,2}^2$ with respect to $s$:

$$\frac{\tilde{\partial} \|\boldsymbol{W}\|_{s,2}^2}{\tilde{\partial} s} = \boldsymbol{W}^2_{\text{least-min}(\text{Dim}(\boldsymbol{W}), \, s + 1)},$$

where $\boldsymbol{W}^2$ is the column-group wise $\mathcal{L}_2$ format of $\boldsymbol{W}$, $\text{Dim}(\boldsymbol{W})$ is the total number of column groups of $\boldsymbol{W}$, and $\boldsymbol{W}^2_{\text{least}-j}$ is the $j$-th least column group in $\boldsymbol{W}^2$.

For a general resource consumption function (e.g., Flops), the non-differentiable part of $R$ is the ceiling function $\lceil s \rceil$, for which we can use a common STE (Bengio et al., 2013): $\frac{\tilde{\partial} \lceil s \rceil}{\tilde{\partial} s} = 1$.

$$\frac{\tilde{\partial} \lceil s \rceil}{\tilde{\partial} s} = 1.$$

Now we can wrap up all the update rules together to be a unified optimization algorithm 1. In this algorithm, we only show an example for the input layer to be column-wise grouped. For a more general case, the column-group wise sparsity can be applied to an arbitrary layer in the RS model.

### A.2 CASCADED PIPELINE AS A BASELINE

We provide experimental results under a cascaded pipeline using **UMEC**, where the compression processes for the input feature and the prediction model are carried out sequentially. Two scenarios are demonstrated: a) Conducting input feature selection first, then prediction model compression, of which the results are shown in Figure 8; b) Conducting embedding dimension reduction first, then prediction model compression, of which the results are shown in Figure 9. We set CR values of first stage in cascaded pipeline as 0.116, 0.116, 0.302, 0.302, 0.302 respectively for a), and 0.1, 0.1, 0.1 for b). Then we try our best to set the final CR values of the whole cascaded pipeline to be similar to the CR values used in our joint framework respectively for fair comparisons in both figures. As an example, we conduct the experimental comparisons on the Criteo AI Labs Ad Kaggle dataset.

Each curve shows the CR-Accuracy trade-off of one compression method. Comparing the results of the cascaded pipeline with the joint framework, Figure 8 and Figuire 9 both support the superiority of the joint framework.

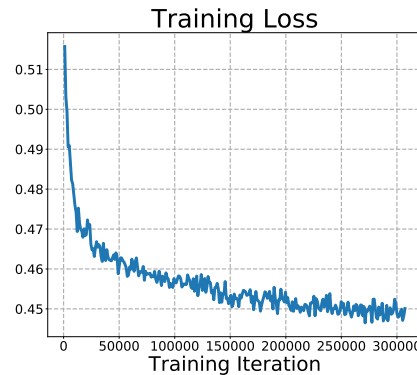

Figure 7: The convergence of the binary cross entropy (BCE) loss during training using **UMEC** with $R_{\text{budget}} = 0.5 \times p_f(\mathcal{W}_0)$, where $p_f(\mathcal{W}_0)$ denotes the Flops of the original dense model.

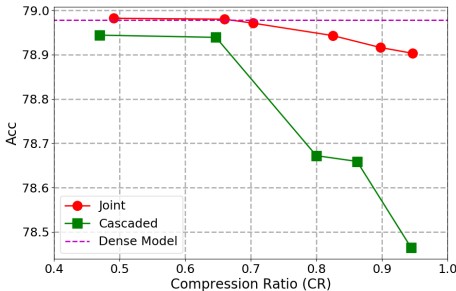

Figure 8: Results on the pipeline with a cascade of input feature selection and prediction model compression, and comparison with the corresponding jointly optimized framework.

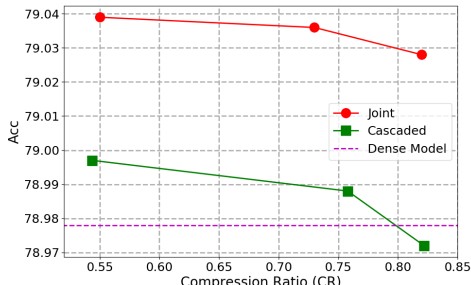

Figure 9: Results on the pipeline with a cascade of embedding dimension reduction and prediction model compression, and comparison with the corresponding jointly optimized framework.

Table 2: Real-device energy consumption and latency of different methods. Best values among all compression methods are shown in bold.

| Method | Acc (%) | Energy ($10^{-3}$J) | Latency (ms) |
|---|---|---|---|
| Ours | **78.971** | **8.81** | **0.123** |
| ECC | 78.939 | 9.05 | 0.132 |
| Handcrafted | 78.963 | 9.32 | 0.140 |
| MP | 78.956 | 9.16 | 0.136 |
| TP | 78.958 | 9.23 | 0.137 |
| Dense Model | 78.978 | 9.49 | 0.144 |

### A.3 ENERGY CONSUMPTION AND INFERENCE LATENCY OF THE NETWORK

We evaluate the energy cost and latency of all models on a GTX 2080 Ti GPU. Following Yang et al. (2019), we use the `nvidia-smi` utility[3] to monitor the energy consumption. We follow the settings in Fu et al. (2020) to measure inference latency on the aforementioned real-device. All experiments are implemented with PyTorch. As shown in Table 2, our method achieves the best accuracy with the least energy consumption and latency among all compared compression methods.

---

[3] https://developer.download.nvidia.com/compute/DCGM/docs/nvidia-smi-367.38.pdf

