# OpenReview forum: "UMEC: Unified model and embedding compression for efficient recommendation systems"
_ICLR.cc/2021/Conference — ICLR 2021 Poster_

### Official Review · AnonReviewer3 · 2020-10-27
**Reviews for UMEC**

**Rating:** 7
**Confidence:** 4

**Review:**

Recommendation models (RM) based on ranking neural networks are nowadays widely adopted in real-world recommendation and retrieval applications. Studying the new problem of RM compression can save energy/latency in those practical systems, and hence has strong application promise.

A typical RM consists of two components: a feature embedding sub-model and a prediction sub-model, both implemented by neural networks. RM compression therefore has to jointly consider the two parts, which makes it different from current CNN/RNN compressions where no input-level compression is needed in general.

To resolve this unique challenge, this paper proposed UMEC, the first unified optimization framework for the recommendation system scenarios. Unlike existing prior arts that treat feature selection and model compression as two separate problems, UMEC jointly learns both of them together, via unifying the original prediction learning goal and the model compression related hard constraints. The optimization is done by first relaxation then ADMM.

Overall this paper is technically sound. The writing is in general good and easy to follow. I have the following questions and suggestions:

The motivation mentions the real latency or power consumption, but the approach uses only FLOPs as the constraint as well as for the evaluation. Is it possible for the authors to report real resource usage such as latency (ms) or energy (J)?

Several acronyms appear before formally defined, such as ADMM (in abstract), BCE (first paragraph), etc. There are also typos like:
-	a RM -> an RM
-	hyper-parameters tuning -> hyper-parameter tuning
-	advertising dataset -> an advertising dataset
-	while manages to -> while managing to
-	and adjust the dimension -> should be “adjusts”

One last nitpick: ADMM was only mentioned in the abstract and the conclusion parts, but not in the method. While I understand that their minimax optimization is essentially ADMM, this may confuse some readers. I suggest the authors to revise their optimization terms for better consistency.

---

> ### Author Response · Authors · 2020-11-21
> **Response to reviewer3**
>
> Thank you very much for appreciating our work and your efforts on paper proofreading.
>
> We report the latency/energy of the compressed model in the newly added table (Table 2) in the appendix.
>
> As for the typos, grammar issues, and terminologies problem, we have modified them in the revised paper.

---

> > ### Comment · AnonReviewer3 · 2020-11-23
> > **Thanks for the rebuttal**
> >
> > The authors have addressed my concerns and I decide to keep my score.

---

### Official Review · AnonReviewer2 · 2020-10-27
**Comments for UMEC**

**Rating:** 7
**Confidence:** 3

**Review:**

1.	Overview:

The paper proposes a new unified optimization framework to solve the RM compression problem. It jointly compresses the prediction network and the feature embedding, and automatically optimizes their complexity budgets under the total resource constraint. In this way, it avoids treating selection of input feature and compression of model as two individual problems, and requires no expensive hyper-parameters tuning for either part. The novel unified model and embedding compression (UMEC) method can directly satisfy both the requirement of resource consumption of the ranking neural network model and the prediction accuracy goal of the ranking model, with end-to-end gradient-based training.

2.	Method:

The overall idea is straightforward, sensible and solidly executed. To ease the hard resource constraint, they first reformulate the inequality constraints as soft regularizations and introduce minimax optimization with dual variables, using the DC (difference of convex functions) representation trick (4). In this way, the importance of regularization terms is automatically tuned by two dual variables in minimax. I find this handling very clever compared to directly tackling a regularized objective (1) with hard-to-tune hyperparameters.

3.	Experiments:

The authors compared with three groups of baselines: model compression, feature selection and embedding reduction. However, I think those are still not enough for showing the proposed joint optimization is indeed superior, by only comparing to each one aspect. The more natural baseline could be to compare with their sequentially cascaded pipeline, yet not end-to-end optimized. For example, what if first running an input feature selection or dimension reduction, then compressing the prediction model (using SOTA methods from each category)

The authors observed that during training, last hidden layer getting the most percentage of neurons removed and the input layer getting the least percentage. I can understand that for binary classification, the last layer can perhaps be less parameterized. But does your observation go against the practice or validity of input feature selection or dimension reduction, i.e. are you suggesting they’re not as useful for compressing RM?

---

> ### Author Response · Authors · 2020-11-21
> **Response to reviewer2**
>
> Thank you very much for your positive feedback on our work.
> 1. Thanks for your good suggestion for choosing the cascaded pipeline as a baseline for comparison. The experiments are still being conducted, and we will report the results as soon as possible before the end of the rebuttal period.  Our preliminary results show that our joint optimization is better than the cascaded baseline. For example, at compression ratios CR1 = 0.35 & CR2 = 0.19, our method gets acc 78.980% & 78.943% while cascaded baseline (conducting input feature selection first, then prediction model compression) has acc 78.939% & 78.672%. More detailed results will be updated in the revised paper soon.
> 2. As for the concern of which part gets mostly compressed, this is the automatic choice of the unified framework and the illustrative result in the paper may be case-specific to that dataset.  The joint optimization allows more flexibility to choose which part to compress. The feature part is more sensitive to final accuracy, so the optimization automatically chooses the easier part to prune, given the compression budget. The case we show in the paper is because the pruning of the hidden layer is already enough to meet the given budget requirement and maintain accuracy. When the budget is very small, the feature number and dimension should both be pruned to help meet the final tradeoff of the budget and accuracy.

---

> ### Author Response · Authors · 2020-11-23
> **Updates on experiments**
>
> Dear Reviewer2,
>
> We have finished the experiments using the cascaded pipeline as a baseline. Please see the full results in appendix A.5 in the latest revised version. If you still have other concerns,  we are glad to discuss them with you. Thanks!

---

### Official Review · AnonReviewer1 · 2020-10-28
**An interesting study on compression of recommendation models**

**Rating:** 7
**Confidence:** 5

**Review:**

This paper studies the compression of recommendation models (RMs). That is new and relatively less studied in the model compression field, but of great practical value. The main unique challenge of RM compression lies in the entanglement of compressing both the network parameters and the feature embedding inputs, and the latter often accounts for more of the computational bottleneck.

To this end, the authors proposed the UMEC framework, by integrating these two sub-tasks into one unified constrained optimization problem, solved by ADMM. Specifically, they develop a resource-constrained optimization that directly sets the target resource consumption and eases the practical usage. The authors conduct extensive experiments and demonstrate the effectiveness of UMEC by observing its superior performance than other state-of-the-art baseline methods. The paper is very well written, and the notations and technical details are clearly presented.

Question 1: My major concern is that, although the authors reported many baseline comparisons and ablation studies, all experiments are on only one dataset (i.e., Criteo AI Labs Ad Kaggle), and one task (CTR prediction). It is unclear whether the proposed method can be generally useful or can be scaled up to industry-level large systems.

Question 2: in Section 4.4 the authors compared with Ginart et al. (2019) for input dimension reduction, while they mentioned another prior work Joglekar et al. (2020) using AutoML to search for feature dimensions. Is it possible to also compare with the later one?

Question 3: in Eqn (1), why only enforcing structured sparsity for the input layer? Wouldn’t it be more natural if also extended to the remaining layers?

---

> ### Author Response · Authors · 2020-11-21
> **Response to reviewer1**
>
> Thank you very much for the positive assessment of our work. We try to answer your insightful questions in detail:
>
> 1: This concern has also been raised by Reviewer 4, so we quote our responses again for your convenience: We are glad to show the performance of our algorithm in other datasets. For demonstration purposes during the short rebuttal period, we choose to perform our training and model compression on a subset dataset from the  Criteo Terabyte Dataset (https://labs.criteo.com/2013/12/download-terabyte-click-logs/), which is the other large scale dataset provided in DLRM benchmark.   We present the compression results over this dataset in the newly added figure in the appendix (Figure 6 and 7), which demonstrates that our proposed method consistently outperforms all other baseline methods. We are glad to deliver the result on the full dataset in the camera-ready version once the paper gets accepted.
>
> 2: Thanks for pointing out this work. This is actually a concurrent NAS work on embedding dimension search, which however hasn’t released the code and searched model architecture yet. Due to the lack of implementation details, we were contacting the authors and will try our best to include comparisons with this work in the final version.
>
> 3: The sparsity for the hidden layers is actually also structured, by setting group size as 1 (input layer’s group size is set to be 16, for serving feature selection purpose.)  and enforcing neuron-wise sparsity, i.e., we prune the entire connections of one neuron, which conforms to the definition of structured pruning.  Yes, we can also extend the group-wise sparsity of the input layer to the hidden layers, but for now, there is no particular need or motivation that the hidden layers should be pruned in a group size larger than 1, in which case the assumption that the hidden layers have a group-wise structure may be too strong and may restrict the searching resolution of the structure causing sub-optimal results.

---

> > ### Comment · AnonReviewer1 · 2020-11-22
> > **Thanks for your response**
> >
> > Thanks for the clarification.  My questions were well addressed. I also read through other reviewers’ comments and responses. In all, I think this is a solid work and I support its acceptance.

---

### Official Review · AnonReviewer4 · 2020-11-05
**Review of UMEC**

**Rating:** 6
**Confidence:** 5

**Review:**

This paper proposes a framework of a unified recommendation system, which balances the compression degree of the model and the accuracy of the model by compressing the embedding layer model, while optimizing feature selection and neural network compression. This paper proposes a resource-constrained method to simultaneously optimize model compression and ensure model accuracy, and use ADMM's solver to optimize constrained models. The experiment verifies the effectiveness of the method by comparing the accuracy of different model compression ratios.

The paper’s key strengths:
1.	This article proposes a constraint optimization method to optimize model learning and model compression at the same time. This method is quite original. And the author proposed computational flops in the article, which defines resource consumption by the number of neuros input and output from the network.
2.	Aiming at the problem of formulated constraint optimization, this paper proposes the use of ADMM to optimize the strategy. And in the appendix of the article, it provides a lot of details about optimization, which is easy to understand.


The paper’s key weaknesses:
1.	The biggest problem of this article is that although it is proposed to optimize feature selection and model selection at the same time to achieve an effect of simultaneously ensuring model accuracy and model compression. But in fact, feature selection itself is performing sparse learning, and the resulting model will become sparse, which has a similar effect to the function of model compression.

2.	In the experimental part, the author compares the accuracy of different methods under different compression ratios to verify the effectiveness of the method. For the experiment in Figure 2, the author showed the effect of compression ratio from 0.5 to 1.0. But I questioned why the author did not show the effect of different methods from 0 to 0.5, because the compression ratio is higher when CR is above 0.5, and there will be more ACC at this time. So I think it might be more meaningful to show the method comparison of CR between 0 and 0.5.

3.	For formula 2, R_budget is a hyperparameter, which represents the parameter of model computation. However, there is no theoretical guarantee and experimental verification of model accuracy and model convergence under different R_budgets. I hope the author will analyze the discussion of a scenario under different R_budget settings.

4.	The experimental scenario in this article is relatively single, and only a set of data is selected for experimental verification. I hope the author can provide more scenarios to verify the effectiveness of the method.

---

> ### Author Response · Authors · 2020-11-21
> **Response to reviewer4**
>
> We sincerely appreciate your valuable comments and thoughtful suggestions. We  address your questions below in details:
>
> 1. Performing feature selection only shrinks the input layer of MLP, and will not affect later hidden layers, which has limited compression ability. Thus, we propose the UMEC framework to unify the feature selection, model (MLP) compression, and embedding dimension reduction in one framework to increase the compression ability to achieve better final compression combinations.
>
> 2. The definition of CR in Section 4.1 denotes the pruned ratio in terms of model FLOPs. If CR equals 0, then no pruning process will be made. If it’s less than 0.5, it’s not easy to distinguish the performance among several compression methods since many of them will behave quite well, and that’s why we leave them out. In Fig 2, we demonstrate several extreme cases where the model gets 90% (CR=0.9), 95% (CR=0.95) pruned in order to show the superiority of our proposed method.
>
> 3. We have already done the experiments under different R_budget settings implicitly, which has been demonstrated in Figure 2, red line. We make R_budgets range from [0.05×pf (W_0) , 0.5×pf (W_0)], which will result in CR ranging from 0.5 to 0.95 when the optimization process finally converges. The number of pruned features/neurons for each layer has been shown in Figure 4 to illustrate the convergence of the proposed algorithm by choosing R_budget = 0.5 × pf (W_0). The convergence demonstration of the training loss is shown in a newly added figure in the appendix.
>
> 4. In the paper, we chose one dataset from DLRM benchmark only to illustrate the efficiency and effectiveness of our proposed framework.  We are glad to show the performance of our algorithm in other datasets.  For demonstration purposes during short rebuttal period, we choose to perform our training and model compression on a subset dataset from the  Criteo Terabyte Dataset (https://labs.criteo.com/2013/12/download-terabyte-click-logs/), which is the other large scale dataset provided in DLRM benchmark.  We present the compression results over this dataset in the newly added figure in the appendix(Figure 6 and 7), which demonstrates that our proposed method consistently outperforms all other baseline methods.  We are glad to deliver the result on the full dataset in the camera-ready version once the paper gets accepted.
>
> If you still have other concerns, we are glad to discuss them with you in detail.

---

> > ### Comment · AnonReviewer4 · 2020-11-25
> > **Response to author**
> >
> > Thanks for the author's response. I think it addressed all my questions and I am glad that the author added additional experiments to demonstrate the effectiveness of the method.

---

### Decision · Program_Chairs · 2021-01-07
**Final Decision**

**Decision:**

Accept (Poster)

**Comment:**

Existing works mostly focus on model compression for the classification task. This paper aims for an efficient recommendation system that can well balance the model compression and model accuracy, which therefore brings in new challenges and opportunities. The authors propose to unify the model compression and feature embedding compression and develop an effective and reasonable solution.  The concerns raised by the reviewers have been well fixed and all reviewers agree on the paper's contribution.  The paper is therefore recommended for acceptance.